# Using Natural Language Processing and Machine Learning to classify the status of kidney allograft in Electronic Medical Records written in Spanish

Andrea Garcia-Lopez[1,2]*, Juliana Cuervo-Rojas[3], Juan Garcia-Lopez[4], Fernando Giron-Luque[2,5]

1 PhD Program in Clinical Epidemiology, Department of Clinical Epidemiology and Biostatistics, Faculty of Medicine, Pontificia Universidad Javeriana, Bogotá, Colombia, 2 Department of Transplant Research, Colombiana de Trasplantes, Bogotá, Colombia, 3 Department of Clinical Epidemiology and Biostatistics, Faculty of Medicine, Pontificia Universidad Javeriana, Bogotá, Colombia, 4 Department of Technology and Informatics, Colombiana de Trasplantes, Bogotá, Colombia, 5 Department of Transplant Surgery, Colombiana de Trasplantes, Bogotá, Colombia

* aegarcia@colombianadetrasplantes.com

## Abstract

### Introduction

Accurate identification of graft loss in Electronic Medical Records of kidney transplant recipients is essential but challenging due to inconsistent and not mandatory International Classification of Diseases (ICD) codes. We developed and validated Natural Language Processing (NLP) and machine learning models to classify the status of kidney allografts in unstructured text in EMRs written in Spanish.

### Methods

We conducted a retrospective cohort of 2712 patients transplanted between July 2008 and January 2023, analyzing 117,566 unstructured medical records. NLP involved text normalization, tokenization, stopwords removal, spell-checking, elimination of low-frequency words and stemming. Data was split in training, validation and test sets. Data balance was performed using undersampling technique. Feature selection was performed using LASSO regression. We developed, validated and tested Logistic Regression, Random Forest, and Neural Networks models using 10-fold cross-validation. Performance metrics included area under the curve, F1 Score, accuracy, sensitivity, specificity, Negative Predictive Value, and Positive Predictive Value.

### Results

The test performance results showed that the Random Forest model achieved the highest AUC (0.98) and F1 score (0.65). However, it had a modest sensitivity (0.76)

**Data availability statement:** R/ We acknowledge your concern regarding data sharing. The structured data after preprocessing from this study is deposited in a public repository. It can be found at: https://www.kaggle.com/ds/6032764 DOI: 10.34740/KAGGLE/DS/6032764.

**Funding:** The author(s) received no specific funding for this work.

**Competing interests:** The authors have declared that no competing interests exist.

and a relatively low PPV (0.56), implying a significant number of false positives. The Neural Network model also performed well with a high AUC (0.98) and reasonable F1 score (0.61), but its PPV (0.49) was lower, indicating more false positives. The Logistic Regression model, while having the lowest AUC (0.91) and F1 score (0.49), showed the highest sensitivity (0.83) with the lowest PPV (0.35).

## Conclusion

We developed and validated three machine learning models combined with NLP techniques for unstructured texts written in Spanish. The models performed well on the validation set but showed modest performance on the test set due to data imbalance. These models could be adapted for clinical practice, though they may require additional manual work due to high false positive rates.

---

## Introduction

Kidney transplantation is the optimal and most effective treatment for patients with end-stage renal disease. The accurate identification in Electronic Medical Records (EMRs) of patients who suffer from graft loss is important for purposes such as disease surveillance, monitoring, risk adjustment, research, evaluation of effectiveness and health care quality improvement. Identifying patients with graft loss in routine clinical records, however, presents a substantial challenge. This is primarily because relevant information may not be directly reflected in codes used for clinical, administrative, billing or insurance claims purposes [1].

Although there is a code for graft loss in the International Classification of Diseases (ICD), it lacks the necessary specificity to fully describe the condition, and its inclusion is not mandatory in most medical records worldwide, resulting in inconsistent and incomplete data. This lack of standardization makes it difficult to accurately identify patients who have lost their grafts. To address the shortcomings of relying solely on ICD codes, the review of medical charts has been identified as the gold standard for the accurate identification of cases [2]. However, the manual review of these records is a labor-intensive endeavor that demands the expertise and time of trained personnel.

In recent years, there has been a significant increase in the use of Natural Language Processing (NLP) techniques and their applications in clinical areas. NLP techniques have been successfully used to identify medical conditions across various clinical disciplines [3–6]. Their application extends beyond traditional algorithms that rely solely on ICD codes. By analyzing textual data from medical records, NLP models can extract relevant information and aid in accurate record classification and case identification.

In the context of kidney transplantation, existing studies have focused on identifying donors with high-risk kidneys [7,8] and developing prediction models of graft loss and mortality by incorporating both structured and unstructured data elements [9]. These studies highlight the potential of using data mining and NLP for data

abstraction. Despite these advances, the application of NLP solutions in kidney transplantation remains in its early stages. Conversely, recent advancements in large language models (LLMs) have significantly transformed the landscape of NLP and artificial intelligence (AI). While generative LLMs have been widely explored for healthcare data analysis, their application in real-world EMR data remains limited due to significant privacy concerns and current challenges such as bias, hallucinations, and impersonal responses [10,11].

The objective of this study was to develop and evaluate NLP and machine learning (ML) models to classify the status of kidney allograft in EMRs written in Spanish.

## Materials and methods

### Data source and study population

In this retrospective cohort study, we included all patients who underwent a kidney transplant between July 2008 and January 2023 at Colombiana de Trasplantes, Colombia. The cohort was composed of 2712 patients with at least one registered medical record after transplantation. A total of 117 566 medical records written in Spanish were available for the study. The data were provided by Colombiana de Trasplantes, one of the largest organ transplantation institutions in Colombia having a network of four transplantation centers across the country. The EMRs data included socio-demographic characteristics, vital signs, paraclinical tests results, prescribed medications, billing codes, and clinical assessments for healthcare services provided within the system. Only unstructured text of outpatient follow-up visits and discharge summaries after hospitalization were included for the study. These records were then merged into one single dataset for analysis. Structured variables and diagnostic codes were not included in the database.

The data were accessed for research purposes on January 22, 2023. Only the primary author (AG-L) could identify individual participants during and after data collection. The other authors did not have access to any information that could identify individual participants. The structured data after preprocessing from this study is deposited in a public repository [12].

### Text mining search

Text mining techniques were utilized to identify cases of renal graft loss from unstructured clinical narratives. The research team systematically searched EMRs for specific keywords and phrases indicative of graft failure, such as "graft loss," "transplant failure," and "loss of kidney function" (listed in Table A.1 in S1 Preprocessing). This approach enabled the extraction and analysis of relevant clinical information and potential cases, which would have been impractical to process manually.

### Manual review to obtain gold standard labels

Graft loss was confirmed through a manual review of all records flagged by the text mining process. This critical phase was carried out by the main author (AG-L), who possesses over five years of research experience in kidney transplantation, following a predefined set of classification criteria (listed in Table A.2 in S1 Preprocessing). These criteria were meticulously established in consultation with clinical experts to ensure accuracy and reliability in the classification process. This collaborative approach with clinical experts provided a robust framework for the manual review, enhancing the validity of the findings. EMRs were classified into two subgroups based on the presence or absence of graft loss ("Yes"/" No").

### Data processing

**Phase 1.** The initial phase of our preprocessing involved text normalization, including converting all text to lowercase, removing dates, punctuation, special characters, blank spaces, numerical digits, and units of measurement. Subsequent tokenization enabled removal of *Stopwords* (common words) and *Custom Stopwords*. Our *Custom Stopwords* list included

proper names, single letters, and units of time, which were identified as irrelevant to the specific objectives of our study (listed in Table A.3 in S1 Preprocessing).

   **Phase 2.** Following the removal of stopwords (common words), medical abbreviations were expanded into full words, as detailed in Table A.4 in S1 Preprocessing. A comprehensive spell-checking process was implemented using an automated spell check tool, which was supplemented by expert supervision (AG-L) to identify and correct errors in context. The expert meticulously reviewed all words to ensure accuracy. The list of original and corrected words can be found in Supplementary File (S2 Reviewed Words).

   This rigorous process revealed various types of spelling errors, which were categorized into accentuation errors, omission errors, substitution errors, and others (Error definition listed in Table A.5 in S1 Preprocessing). All identified errors were corrected within the text, and in cases in which the words were not meaningful in context (less than 0.1%), they were replaced with placeholders. This dual approach ensured that the corrections were not only accurate but also appropriate for the text's meaning and usage. Custom stopwords were removed as the final step of Phase 2 of data processing.

   **Phase 3.** Subsequently, we addressed the issue of low-frequency words. Words that appeared fewer than 10 times across all records were eliminated. The cutoff of 10 occurrences was chosen arbitrarily, based on preprocessing findings, and these words were deemed to have low clinical relevance. Lastly, the preprocessing phase included the removal of duplicate spaces within the text.

## Natural Language Processing

We developed and evaluated a model aimed at identifying graft loss and classifying records, employing NLP techniques facilitated by the "tm" R package (v 0.7.11). Initially, we constructed a Document-Term Matrix (DTM) to convert the text data into a structured, numerical format conducive to analysis by machine learning algorithms. The stemming process was used to reduce the complexity of the text data consolidating variations of a word into a single representative form. N-grams were generated after the tokenization process. To refine our dataset further, we utilized the "removeSparseTerms" function, effectively eliminating sparse terms that contribute minimal informational value to the analysis.

## Data partition

The dataset was initially partitioned using an 80/20 split, where 80% of the data was allocated for training and validation, and the remaining 20% was reserved for testing. The training and validation set underwent a 10-fold cross-validation process. In this method, the data was divided into 10 equal folds, with 9 folds used for training and 1-fold used for validation in each iteration. This cross-validation process was repeated 10 times, ensuring that each fold was used as the validation set exactly once. The final model performance was evaluated on the test set, which comprised the remaining 20% of the data.

## Feature selection

Feature selection was performed on the training set using two techniques: bivariate analysis with a p-value threshold of ≤ 0.01 for a statistically significant association with graft loss, and LASSO (least absolute shrinkage and selection operator) selecting the highest coefficients. This meticulous preprocessing resulted in a polished Document Term Matrix (DTM), which formed the foundation of our final analytical process.

## Balancing the dataset

To balance the dataset, we employed the undersampling technique, which involves randomly reducing the number of instances in the majority class. This approach can enhance the performance of classification algorithms by ensuring that the model does not become biased towards the majority class and permits a more valid evaluation of accuracy and

precision. The balancing was performed on the training set before applying cross-validation, while the test set remained imbalanced to evaluate the model in a real-world scenario. We use the "ROSE" (Random Over-Sampling Examples) library in R [13].

### Model development and validation

Three distinct supervised ML models were trained, validated, and tested: Logistic Regression, Random Forest, and Neural Networks. Each one was customized with unique parameters without using hyperparameter optimization techniques, leading to the evaluation of a total of 15 models. The models were trained on 9-folds of the cross-validation, validated on 1-fold with different hyperparameters, and tested on the imbalanced test set to evaluate the model in a real-world scenario. The performance of each NLP algorithm was compared to the gold standard manually annotated labels of graft loss. We selected the top three algorithms from each class—Logistic Regression, Random Forest, and Neural Network—for comparison. We used the following performance metrics to evaluate the models: AUC (Area Under the Curve), F1 Score, accuracy, sensitivity, specificity, negative predictive value (NPV), and positive predictive value (PPV). All analyses were performed using R Statistical Software (v4.0.3; R Core Team 2020). Fig 1 shows an overview of the methods.

This study received approval from the Research and Ethics Committee of the Faculty of Medicine of Pontificia Universidad Javeriana (REF 266/2021), ensuring adherence to national and international ethical standards.

## Results

### Characteristics of extracted documents

Among the 2712 patients, graft loss occurred in 374 individuals. Most of the patients had more than one medical record post transplantation; the average number of records per patient was 43. The final corpus contained 117 566 documents, of which 1063 (0.9%) were records pertaining to 374 individuals experiencing graft loss. Among all documents, 98.6% were outpatient notes with the remaining 1.32% being discharge summaries. Documents associated with patients experiencing graft loss exhibited fewer words overall, fewer number of words appearing only once (frequency n=1), a lower average number of notes per patient, and a higher average number of words per note compared to documents from patients without graft loss. Detailed statistics of the characteristics of documents can be found in Table 1 and characteristics of words across documents can be found in Table 2.

### Key concepts

The word cloud visualization in Fig 2 provides a visual summary of common words found in documents related to patients with graft loss. Terms such as "creatinine" (creatinina) "nephrectomy" (nefrectomia), and "hemodialysis" (hemodiálisis) appear with higher occurrence or significance than others within the dataset.

Fig 3 depicts a semantic network that maps out the complex interconnections among various medical terms. These terms extracted from the EMRs of kidney transplant patients represent those who have experienced graft loss. Whitin this network, several notable associations have been identified, including dysfunction-transplant (disfunción-trasplante), dysfunction-chronic (disfunción-crónica), loss-graft (pérdida-injerto), unit-renal (unidad-renal), nephrectomy-graft (nefrectomia-injerto) and, nephrectomy-transplant (nefrectomia-trasplante). These associations are found not only to be clinically relevant, but also to accurately represent the realities of clinical practice.

### Class distribution

Before undersampling the dataset was highly imbalanced with 99.1% of the records corresponding to instances without graft loss and with only 0.9% of them corresponding to the outcome of interest. After applying undersampling in the training set, the distribution was balanced to 50% for each class (records without graft loss = 834; records with graft loss = 846).

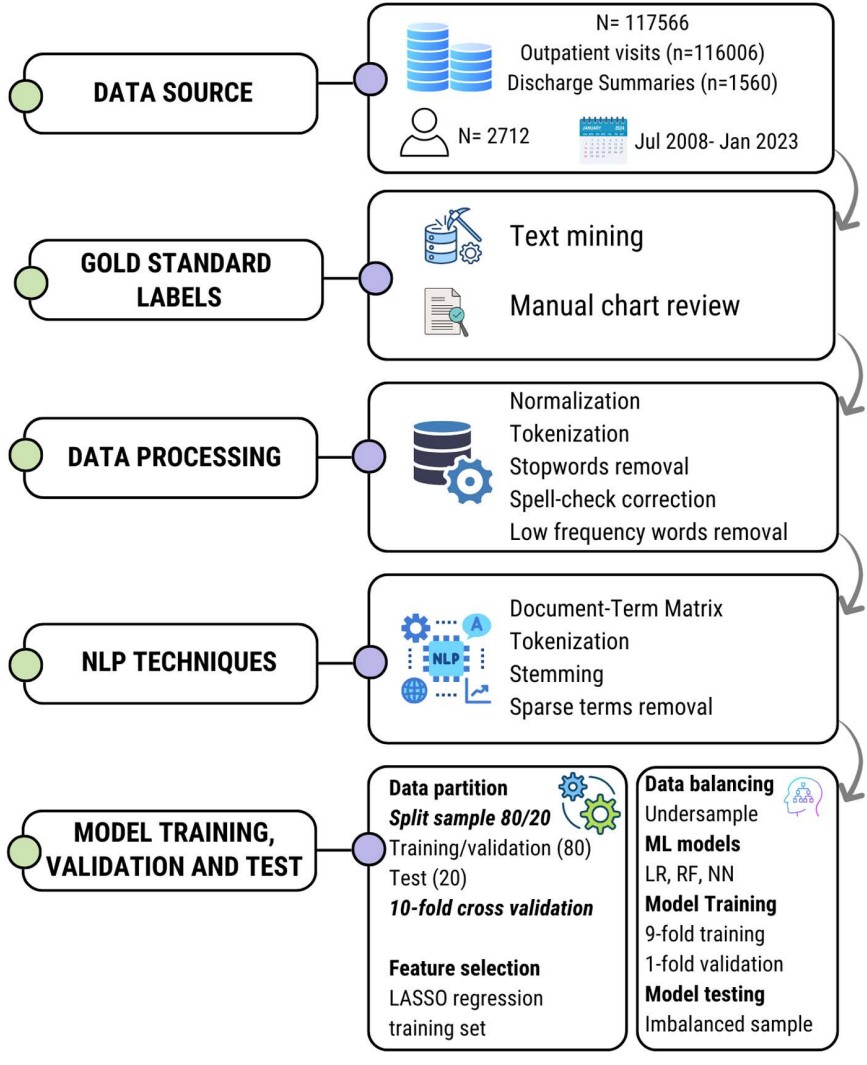

**Fig 1. Overview of the methods.** NLP, Natural language processing; ML, Machine learning.

**Table 1. Document characteristics.**

| Characteristics of medical records | All (n = 117566) | Records with graft loss reported (n = 1063) | Records without graft loss reported (n = 116503) |
|---|---|---|---|
| **Number of medical records** | | | |
| Number of Discharge summaries (%) | 1560 (1.32%) | 103 (9.6%) | 1457 (1.2%) |
| Number of Outpatients notes (%) | 116006 (98.6%) | 960 (90.3%) | 115046 (98.7%) |
| **Average number of records per patient (Min-Max)** | 43.3 (1-206) | 2.8 (1-70) | 43.5 (1-204) |

**Table 2. Word characteristics.**

| Characteristics of words accross documents | Total word count (n = 11017383) | Word count in records with graft loss reported (n = 136357) | Word count in records without graft loss reported (n = 10881026) |
|---|---|---|---|
| **Words appearing only once (frequency n = 1) (%)** | 65202 (0.59%) | 5233 (3.8%) | 64560 (0.59%) |
| **Word count after preprocessing (%)** | 6482026 (100%) | 79503 (1.3%) | 6402523 (98.7%) |
| **Count of distinct words across the documents** | | | |
| Before preprocessing | 123333 | 10174 | 122186 |
| After preprocessing | 9036 | 4455 | 9034 |
| **Average number of words per record (Min-Max)** | | | |
| Before preprocessing | 93.7 (5-683) | 128.2 (11-559) | 93.3 (5-683) |
| After preprocessing | 55.1 (1-468) | 74.7 (5-309) | 54.9 (1-475) |

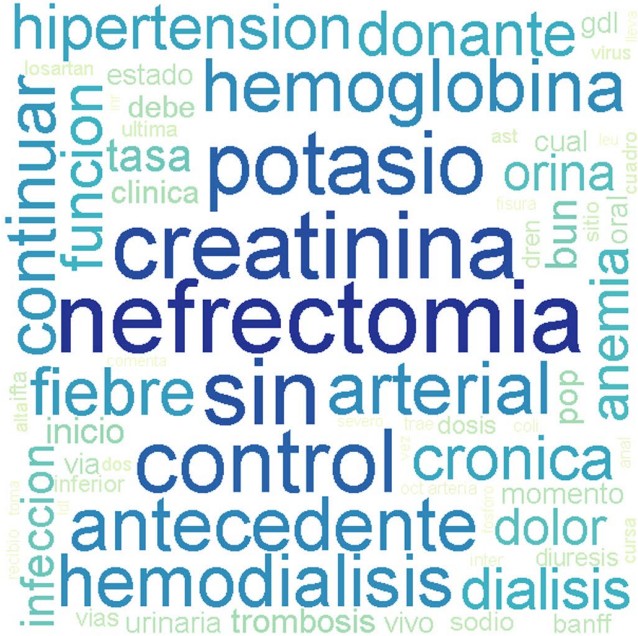

**Fig 2. Word cloud visualization represents the frequency of key terms extracted from the dataset of patients with graft loss.** Larger and bolder words like "creatinina" (creatinine), "nefrectomia" (nephrectomy), indicate their higher occurrence or significance within the dataset.

Class distribution on the test set was 23296 (99.08%) for records without graft loss and 217 (0.92%) for records with graft loss.

## Selected features

The DTM initially contained 612 variables. After performing a bivariate analysis, 301 variables were retained based on their statistically significant association with graft loss (p-value ≤ 0.01). Subsequently, the LASSO technique was applied, which further reduced the number of features to six (nefrectomía, pérdida, hemodiálisis, unidad, terapia, diálisis). These six features were then used to train the models.

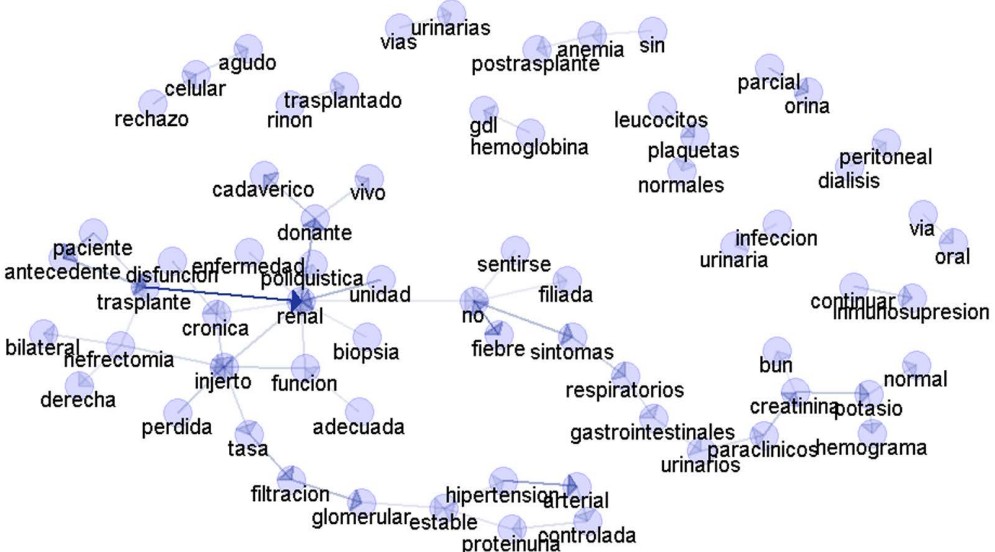

**Fig 3. The image depicts a semantic network illustrating the relationships between various medical terms from the dataset of patients with graft loss.** The nodes represent specific terms such as "transplanted," "kidney," "donor," "biopsy," "function," "infection," and "dialysis," among others. The edges between nodes indicate the co-occurrence or correlation between these terms within the context of post-transplant patient management and outcomes.

## Model performance

**Validation performance.** The performance of each NLP algorithm was compared to the gold standard of manually annotated labels of graft loss. We selected the top three algorithms from each class—Logistic Regression, Random Forest, and Neural Network—for comparison. The validation performance metrics, reported as means and standard deviations (SD) of results from 10-fold cross-validation, indicated that Random Forest and Neural Network models achieved the highest mean AUC of 0.98 (0.00), compared to Logistic Regression's 0.95 (0.01). Logistic Regression, however, demonstrated superior mean F1 score of 0.95 (0.01) and mean sensitivity of 0.94 (0.02), suggesting a better balance between precision and recall and a higher true positive rate. In terms of specificity and positive predictive value (PPV), Random Forest excelled with mean values of 0.99 (0.00) and 0.99 (0.00), respectively, indicating its robustness in identifying true negatives (Table 3. Shows performance metrics of the top three classification models of each class).

**Test performance.** When evaluating test performance, Random Forest and Neural Network models maintained a high AUC of 0.98, outperforming Logistic Regression's AUC of 0.91. Random Forest also achieved the highest F1 score of 0.65, reflecting a better balance between precision and recall on the test set. Both Random Forest and Neural Network models demonstrated superior accuracy (0.99) and specificity (0.99), compared to Logistic Regression's accuracy of 0.98 and specificity of 0.98. Despite Logistic Regression's higher sensitivity of 0.83, Random Forest and Neural Network models showed better overall performance in terms of PPV, with values of 0.56 and 0.49, respectively. All three models exhibited an equally high negative predictive value (NPV) of 0.99, indicating their effectiveness in identifying negative cases correctly on the test set.

Based on the results, the Random Forest model emerged as the best performing. This model achieved the best F1 score (0.65) on the test set, providing a better balance between precision and recall compared to the other models. This balance is crucial for ensuring that the model performs well in both identifying true positives and minimizing false positives (see Table 3).

Confusion Matrix, AUC and the code of all models are provided in Supplementary File (S3 Model's Performance).

**Table 3. Performance metrics of the top three classification models of each class.**

| Performance Metrics | Models | | |
|---|---|---|---|
| | Logistic Regression | Random Forest | Neural Network |
| **Validation performance**[a] | | | |
| Mean AUC (SD) | 0.95 (0.01) | 0.98 (0.00) | 0.98 (0.00) |
| Mean F1 (SD) | 0.95 (0.01) | 0.84 (0.02) | 0.91 (0.01) |
| Mean Accuracy (SD) | 0.92 (0.03) | 0.86 (0.01) | 0.91 (0.01) |
| Mean Sensitivity (SD) | 0.94 (0.02) | 0.73 (0.03) | 0.84 (0.03) |
| Mean Specificity (SD) | 0.96 (0.01) | 0.99 (0.00) | 0.98 (0.01) |
| Mean PPV (SD) | 0.96 (0.01) | 0.99 (0.00) | 0.98 (0.01) |
| Mean NPV (SD) | 0.94 (0.02) | 0.78 (0.02) | 0.86 (0.02) |
| **Test performance** | | | |
| AUC | 0.91 | 0.98 | 0.98 |
| F1 | 0.49 | 0.65 | 0.61 |
| Accuracy | 0.98 | 0.99 | 0.99 |
| Sensitivity | 0.83 | 0.76 | 0.81 |
| Specificity | 0.98 | 0.99 | 0.99 |
| PPV | 0.35 | 0.56 | 0.49 |
| NPV | 0.99 | 0.99 | 0.99 |

AUC: area under the curve; SD: Standard deviation; PPV: Positive predictive value; NPV: Negative predictive value.

[a]Validation performance metrics are reported as means and SD of results of a 10-fold cross validation.

Fig 4 comprises two key components that elucidate the performance and characteristics of the Random Forest algorithm. Fig 4a presents the Confusion Matrix with a comprehensive overview of the model's performance metrics on the test set. Fig 4b shows the Variable Importance plot and highlights the variables that contribute most significantly to the model's predictions. This plot ranks the variables based on their importance scores.

## Discussion

In this study, we introduce novel NLP algorithms to classify the status of kidney allografts using EMRs written in Spanish. The significance of our results lies in showing that these NLP algorithms achieve sufficient accuracy, indicating their potential for deployment not only in research settings but also in clinical applications. In this section, we discuss the key findings of the study, the implications of these results, and the limitations of this work.

## Key findings

This study found that NLP and ML models can accurately classify the status of kidney allografts in EMRs in unstructured text written in Spanish. Among the models evaluated, all showed strong performance on the balanced validation set. However, when tested on the imbalanced test set, which represents a real-world scenario, all models exhibited a decrease in performance. This decline is likely due to the sample imbalance, which can hinder the model's ability to accurately predict minority class instances. Despite this, the Random Forest model outperformed the others achieving an AUC of 0.98 and an F1 score of 0.65, indicating strong discrimination between EMRs that recorded graft loss and those that did not, as well as a reasonable balance between precision and sensitivity. Notably, the model showed a low PPV (0.56) and modest sensitivity (0.76), indicating that a significant number of predicted positives are false positives, which implies additional manual work to verify these cases. Addressing this limitation is crucial for enhancing the practical applicability of our model in clinical settings.

A

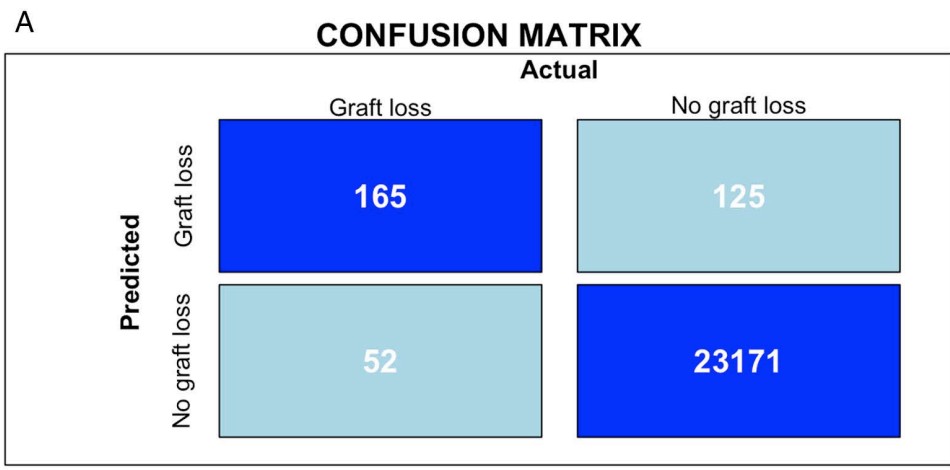

**CONFUSION MATRIX**

**VARIABLE IMPORTANCE**

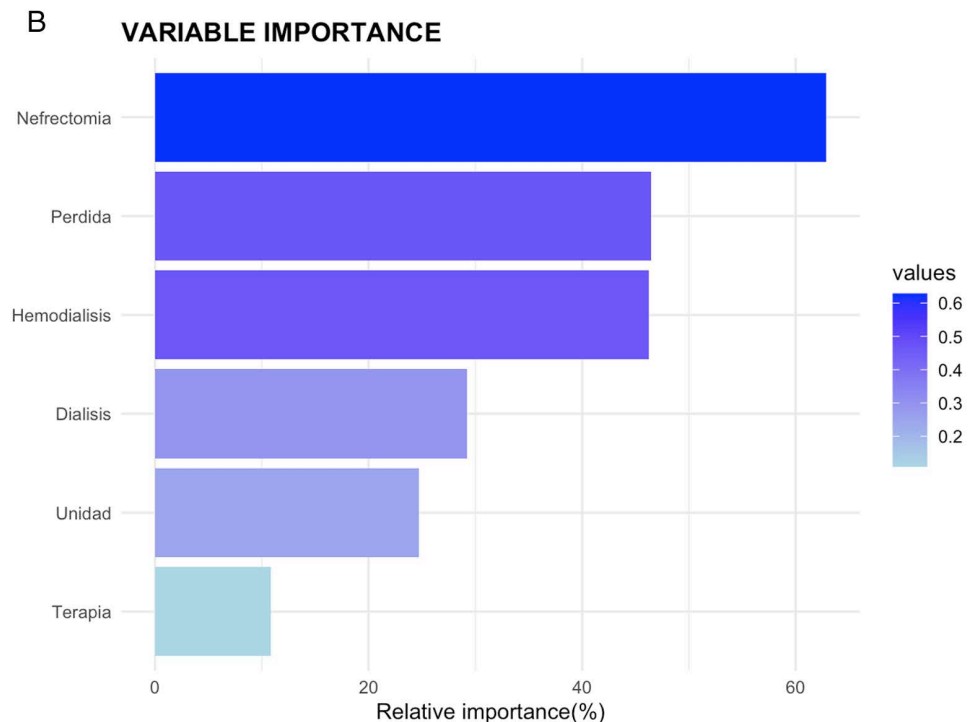

**Fig 4. a) Confusion Matrix showing a comprehensive overview of random model performance metrics; b) Variable importance plot highlights the variables that contribute most significantly to the model's predictions based on their importance scores.**

Neural Networks have gained significant attention in recent times due to their ability to capture complex patterns. However, it's important to note that they necessitate a substantial volume of data for effective training and excel in scenarios characterized by highly non-linear relationships [14,15].

## Comparison with other models

To our knowledge this is the first NLP algorithm introduced to classify the status of kidney allograft in transplant patients based on unstructured EMRs in Spanish language. However, NLP has been successfully applied across different medical domains to identify conditions from clinical narratives. For instance, Jie Pan and colleagues leveraged EMRs to detect cerebrovascular disease [16]. Their NLP algorithm outperformed the traditional ICD code algorithm in terms of both sensitivity (70% vs. 25%) and PPV (88% vs. 83%). Similarly, Ali Ebrahimi and his team developed an NLP pipeline alongside machine learning models to accurately classify patients' smoking status from EMRs, achieving exemplary classification performance with an F1 score of 98%, PPV of 99%, and sensitivity of 98% [17]. Furthermore, Nakeya Dewaswala and collaborators successfully extracted diagnoses of hypertrophic cardiomyopathy from cardiac magnetic resonance narrative texts using an NLP system, which reported an accuracy of 0.99 [3]. These studies show the utility and potential of NLP tools in transforming unstructured clinical text into structured data, to extract meaningful information and facilitate the identification of outcomes or medical conditions reducing the time and effort required for these tasks.

The application of NLP algorithms in the context of kidney transplantation has primarily been focused on predicting outcomes rather than classifying the status of the kidney allograft based on EMRs [8,18]. This is where the novel application of NLP for classification purposes, as described in our study, becomes relevant, introducing a new approach to address the shortcomings of relying on ICD codes or manual review for the accurate identification of cases in medical or administrative records.

When comparing our NLP algorithms to other state-of-the-art techniques in similar domains, our models showed considerable advantages in terms of discrimination, calibration and accuracy. However, while these models can be useful in clinical practice, manual work may still be required due to the high number of false positives. Moreover, a distinctive feature of our models is their application to Spanish-language texts. This differentiates them from many existing NLP tools which are predominantly designed for English-language text [19,20].

## Implications

Several studies have shown that NLP algorithms outperform traditional ICD coding in accurately identifying clinical conditions [21–24]. Although there is an ICD code for graft loss, its entry is not mandatory in most medical records globally, leading to inconsistent and incomplete data. This lack of standardization complicates the accurate identification of patients who have lost their grafts. This inconsistency and incompleteness of data is the primary reason why we did not rely on ICD codes for this study. Implementing these NLP models using EMRs can significantly enhance the surveillance and monitoring of outcomes in kidney transplant patients, enabling faster and more accurate identification of graft loss cases.

To implement the model in real-world scenarios, several improvements are necessary. Despite the promising results, the model may still require manual assistance due to the high number of false positives to ensure accuracy and reliability. This is particularly important given the variability and complexity of real-world data, which can differ significantly from the controlled conditions of a study. The current model can be used to support text review, aiding clinicians and researchers in identifying cases of graft loss more efficiently.

In the future, the model can be enhanced with the integration of large language models (LLMs). LLMs have shown significant advancements in natural language processing (NLP) and can improve the model's ability to understand and process complex medical texts [25,26]. This enhancement could lead to more accurate classifications and a reduction in the need for manual intervention. Additionally, continuous training and validation with diverse datasets will be crucial to ensure the model's robustness and generalizability across different clinical settings.

Additionally, by developing and validating these NLP algorithms in Spanish language, our study provides a useful resource for healthcare providers in Spanish-speaking countries. This contribution facilitates more accurate and efficient classification of kidney allograft status, addressing a significant gap in the availability of tailored medical technologies for non-English-speaking regions [19,20].

## Strengths and limitations

This study has several notable strengths. One of the most significant is its large sample size, which enhances the statistical reliability and robustness of the findings. Additionally, the use of robust methods further strengthens the validity of our results. Importantly, a meticulous preprocessing stage was undertaken in this study. This rigorous preprocessing could have significantly improved the performance of the models, as it ensured that the data fed into the models was clean, relevant, and well-structured.

Despite its promising results, this study is subject to several limitations. Firstly, our model's performance is contingent upon the quality and specificity of the input data, which may vary across different healthcare systems. Such variability could impair the model's performance when deployed outside the controlled parameters of this study, underscoring the necessity for continual testing, validation and refinement in new environments. Secondly, the model's performance is compromised by its high weighting of the term "nephrectomy," which, despite its importance, appears infrequently in graft loss cases. Thirdly, since the study is based on data from a single institution in Colombia, extrapolating the results to other institutions or countries may be challenging due to variations in clinical practices and documentation standards. Lastly, the NLP algorithms developed for this study are specifically designed for Spanish-language EMRs, limiting their immediate applicability to other languages and healthcare systems. Adapting these models to other languages would necessitate further validation and training on relevant datasets, which may not be immediately accessible. Additionally, the model's classification ability decreases in scenarios with highly imbalanced data. However, it can still be used as clinical support or assistance. This highlights the importance of continuous improvement and adaptation of the model to ensure its effectiveness in real-world applications. Furthermore, hyperparameter optimization techniques, which could significantly enhance model performance, were not utilized due to the high computational demands, making it unfeasible for this particular study.

## Future work

Large Language Models represent a significant advancement over traditional NLP methods due to their ability to understand context, generalize across tasks, leverage transfer learning, and continuously improve [27]. These characteristics make them especially powerful for complex and nuanced language tasks, such as those encountered in the analysis of EMRs and other sophisticated text processing applications. We will explore the potential of Large Language Models for future work to further enhance their application in these and other domains.

## Conclusion

We present the results of a novel model capable of classifying the status of kidney allografts using unstructured text in EMRs written in Spanish. By combining NLP methods with a Random Forest algorithm our model achieved a moderate level of classification performance with relatively low levels of error in the detection of records reporting graft loss. This algorithm could be utilized to enhance and complement identification through existing ICD coding, contributing to identify cases in massive databases of unstructured text not amenable to manual review and classification, which may support the use of untapped health services data for research and epidemiological surveillance.

## Supporting information

**S1 Preprocessing. Supporting information of data cleaning and preprocessing.**
(DOCX)

**S2 Reviewed Words.** List of original and corrected words after spell checking. (XLS)

**S3 Model's Performance.** Code and output of logistic regression, random forest and neural networks models. (HTML)

## Acknowledgments

We would like to thank Colombiana de Trasplantes for their support and for providing the necessary data for this study. The author would like to thank the many members of the Javeriana Clinical Epidemiology PhD Program for their constructive discussions related to the topic of this paper.

## Author contributions

**Conceptualization:** Andrea Garcia-Lopez, Juliana Cuervo-Rojas, Juan Garcia-Lopez, Fernando Giron-Luque.

**Data curation:** Andrea Garcia-Lopez, Juan Garcia-Lopez.

**Formal analysis:** Andrea Garcia-Lopez, Juan Garcia-Lopez.

**Investigation:** Andrea Garcia-Lopez, Juliana Cuervo-Rojas, Fernando Giron-Luque.

**Methodology:** Andrea Garcia-Lopez, Juliana Cuervo-Rojas.

**Project administration:** Fernando Giron-Luque.

**Supervision:** Andrea Garcia-Lopez, Juliana Cuervo-Rojas, Fernando Giron-Luque.

**Validation:** Andrea Garcia-Lopez, Juliana Cuervo-Rojas, Fernando Giron-Luque.

**Visualization:** Andrea Garcia-Lopez.

**Writing – original draft:** Andrea Garcia-Lopez.

**Writing – review & editing:** Andrea Garcia-Lopez, Juliana Cuervo-Rojas, Juan Garcia-Lopez, Fernando Giron-Luque.

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
