## [Decision Letter · Decision Letter 0]

25 Sep 2024

PONE-D-24-30503Using Natural Language Processing and Machine Learning to classify the status of kidney allograft in Electronic Medical Records written in SpanishPLOS ONE

Dear Dr. Garcia-Lopez,

Thank you for submitting your manuscript to PLOS ONE. After careful consideration, we feel that it has merit but does not fully meet PLOS ONE’s publication criteria as it currently stands. Therefore, we invite you to submit a revised version of the manuscript that addresses the points raised during the review process. Please note that submitting a revised manuscript does not necessarily guarantee that this article will be eventually accepted. The reviewer has diligently highlighted several important aspects of this study that substantially limit the veracity of the study's findings.

We look forward to receiving your revised manuscript.

Kind regards,

Abdulaziz T. Bako, MBBS; MPH; PhD

Academic Editor

PLOS ONE

Journal Requirements:

4. In the online submission form, you indicated that [Data cannot be shared publicly because data are owned by Colombiana de Trasplantes. Data are available upon request for researchers who meet the criteria for access to confidential data (contact the main author via email).]. 

5. We note that Figure 1 in your submission contain copyrighted images. All PLOS content is published under the Creative Commons Attribution License (CC BY 4.0), which means that the manuscript, images, and Supporting Information files will be freely available online, and any third party is permitted to access, download, copy, distribute, and use these materials in any way, even commercially, with proper attribution. For more information, see our copyright guidelines: http://journals.plos.org/plosone/s/licenses-and-copyright.

Reviewers' comments:

Reviewer's Responses to Questions

**Comments to the Author**

1. Is the manuscript technically sound, and do the data support the conclusions?

Reviewer #1: Partly

2. Has the statistical analysis been performed appropriately and rigorously? 

Reviewer #1: No

3. Have the authors made all data underlying the findings in their manuscript fully available?

Reviewer #1: No

4. Is the manuscript presented in an intelligible fashion and written in standard English?

Reviewer #1: Yes

5. Review Comments to the Author

Reviewer #1: The article reports a method for identifying graft loss in electronic medical records. It highlights the creation of a database of 2712 patients with a total of 117566 unstructured medical records. A text feature extraction scheme based on NLP by means of word and N-gram identification and the use of machine learning models for a binary classification scheme is reported.

Although there is an interesting contribution in the creation of the database and the application in medical records related to kidney transplantation, there are several methodological problems in the study, and the contribution from the technique is not clear.

The major findings are described below:

The references cited in the introduction regarding the use of NLP in the identification of medical conditions are not state of the art. The most current one is from 2022 and does not refer to the use of large language models, which is what is currently used.

The methods used for word count and N-grams do not take context into account. For example, it does not take into account whether a term is present, absent or a possibility. This changes the interpretation and can affect the result. On the other hand, the methodology used may depend on whether the terms have typographical errors (which is common in medical records). Although a method of spelling correction is reported, it is not optimal.

It is not clear whether the results presented are from a test set. In Table 3, if the results are from cross-validation they should be presented with standard deviation. When reviewing the confusion matrix, it appears that all medical records were used. On the other hand, it is observed that the high specificity and high accuracy reported is an effect of class imbalance. This is verified when it is seen that the number of false positives is similar to the true positives, giving an accuracy of 0.519. The most important measure for the analysis should be the F1 score, which is reported to be 0.651 (Figure 4.a).

When reviewing the confusion matrices of the supplementary material, it is not understood why for some models the matrix is reported with 1063 positive cases or other models with 101 positive cases and others with 119.

Other suggestions:

The manual labels were made by a single specialist. At least two specialists are suggested to perform concordance measurements.

6. PLOS authors have the option to publish the peer review history of their article (what does this mean? ). If published, this will include your full peer review and any attached files.

**Do you want your identity to be public for this peer review?** For information about this choice, including consent withdrawal, please see our Privacy Policy .

Reviewer #1: No

---

## [Author Response · Author response to Decision Letter 1]

8 Nov 2024

'Response to Reviewers'

Journal requirements

If applicable, we recommend that you deposit your laboratory protocols in protocols.io to enhance the reproducibility of your results.

R/ Not applicable, our protocol does not pertain to laboratory procedures.

R/ We have thoroughly reviewed our manuscript and confirm that it adheres to all of PLOS ONE’s style requirements.

Please note that PLOS ONE has specific guidelines on code sharing for submissions in which author-generated code underpins the findings in the manuscript. In these cases, we expect all author-generated code to be made available without restrictions upon publication of the work. Please review our guidelines at https://journals.plos.org/plosone/s/materials-and-software-sharing#loc-sharing-code and ensure that your code is shared in a way that follows best practice and facilitates reproducibility and reuse.

R/ To comply with these guidelines, we shared the code in a public repository. This will allow other researchers to access, reproduce, and build upon our work, thereby enhancing the transparency and impact of our research.

Code for models and validation can be found at:

https://www.kaggle.com/code/aegarcialo/data-balanced-logistic-regression-and-random-fore

It is also attached in Supplemental material (S3 and S4)

We note that you have indicated that there are restrictions to data sharing for this study. PLOS only allows data to be available upon request if there are legal or ethical restrictions on sharing data publicly. For more information on unacceptable data access restrictions, please see http://journals.plos.org/plosone/s/data-availability#loc-unacceptable-data-access-restrictions

R/ We acknowledge your concern regarding data sharing. The structured data after preprocessing from this study is deposited in a public repository.

It can be found at:

https://www.kaggle.com/ds/6032764

DOI: 10.34740/KAGGLE/DS/6032764

The original clinical data with unstructured text will be available upon request to the main author via email

R/ We acknowledge your request regarding the figures. The image presented in the previous version of the paper was our own design, but it was created using a free version of a program, which resulted in watermarks. We have now replaced Figure 1 with a new version (created by the authors), ensuring that the design was created and downloaded using the Pro version of the program (Pro content license). This allows us to use and publish the elements without any watermarks.

The image was designed (by the authors) using Canva Pro, which grants non-exclusive licenses to use content under the Pro Content License (also known as One Design Use License). This allows us to use and publish the elements of the design in accordance with the license terms, ensuring that all components of the design comply with usage requirements.

More information can be found here: https://www.canva.com/policies/content-license-agreement/

Comments to the Author

Review Comments to the Author

Reviewer #1: The article reports a method for identifying graft loss in electronic medical records. It highlights the creation of a database of 2712 patients with a total of 117566 unstructured medical records. A text feature extraction scheme based on NLP by means of word and N-gram identification and the use of machine learning models for a binary classification scheme is reported.

Although there is an interesting contribution in the creation of the database and the application in medical records related to kidney transplantation, there are several methodological problems in the study, and the contribution from the technique is not clear.

The major findings are described below:

Comment:

- The references cited in the introduction regarding the use of NLP in the identification of medical conditions are not state of the art. The most current one is from 2022 and does not refer to the use of large language models, which is what is currently used.

R/ Thank you for your feedback. We have updated the references in the introduction to include the most recent studies, particularly those focusing on the use of large language models in the identification of medical conditions.

We also added the following text in the introduction: “Conversely, recent advancements in large language models (LLMs) have significantly transformed the landscape of NLP and artificial intelligence (AI). While generative LLMs have been widely explored for healthcare data analysis, their application in real-world EMR data remains limited due to significant privacy concerns and current challenges such as bias, hallucinations, and impersonal responses (10,11)”.

Comment:

-The methods used for word count and N-grams do not take context into account. For example, it does not take into account whether a term is present, absent or a possibility. This changes the interpretation and can affect the result. On the other hand, the methodology used may depend on whether the terms have typographical errors (which is common in medical records). Although a method of spelling correction is reported, it is not optimal.

R/. We acknowledge that the methods used for word count and N-grams do not fully account for context, which can indeed influence the interpretation and results. On the other hand, we have conducted a thorough spell check and analyzed the context word by word to address these issues. We added the following text in the methods section: “Following the removal of stopwords, medical abbreviations were expanded into full words, as detailed in S1 Table A.4. A comprehensive spell-checking process was implemented using an automated spell check tool, which was supplemented by expert supervision (AG-L) to identify and correct errors in context. The expert meticulously reviewed all words to ensure accuracy. The list of original and corrected words can be found in Supplementary File (S2).

This rigorous process revealed various types of spelling errors, which were categorized into accentuation errors, omission errors, substitution errors, and others (Error definition listed in S1 Table A.5). All identified errors were corrected within the text, and in cases where the words had no meaningful context (less than 0.1%), they were replaced with placeholders”.

We believe that with these corrections and the detailed context analysis, it is possible to bypass the need for analyzing whether terms are present, absent, or possibilities. By ensuring the accuracy and context of each word, we have mitigated the potential misinterpretations that could arise from typographical errors or lack of meaning in context.

Comment

-It is not clear whether the results presented are from a test set. In Table 3, if the results are from cross-validation they should be presented with standard deviation. -When reviewing the confusion matrix, it appears that all medical records were used.

R/ Thank you for your observation. In our previous analysis, some results were indeed analyzed using out-of-bag (OOB) estimates of some Random Forest models. However, we have implemented sample balancing and re-evaluated all models using the test set for evaluation.

We have updated Table 3 to reflect these changes, ensuring that all results are now derived from the balanced test set. Additionally, for results obtained through cross-validation, we have included the standard deviation of the measure of overall accuracy to provide a clearer understanding of the variability in our overall performance metric.

Comment

-On the other hand, it is observed that the high specificity and high accuracy reported is an effect of class imbalance. This is verified when it is seen that the number of false positives is similar to the true positives, giving an accuracy of 0.519.

R/We acknowledge that the high specificity and accuracy initially reported were influenced by class imbalance. To address this, we have implemented sample balancing using the under-sampling technique on the registries without graft loss, which constituted the majority, and re-evaluated all models with the balanced sample.

The updated results, derived from the balanced dataset, provide a more accurate representation of model performance. Table 3 reflect these changes and the code and output is provided in supplemental material S3 and S4.

It is important to notice that even in this balanced dataset all our models present relatively high performance on all metrics, with values of accuracy and F1 scores above 0.93.

We appreciate your feedback and believe these adjustments were conducive to a more rigorous evaluation of model performance, and thus enhance the validity and reliability of our findings.

Comment

-The most important measure for the analysis should be the F1 score, which is reported to be 0.651 (Figure 4.a).

R/ We agree that the F1 score is a crucial measure for evaluating model performance. In response to your comment, we have included the F1 score as one of the key metrics in the analysis of all models. After balancing the sample, the F1 score improved substantially, and now ranges between 0.935 and 0.955

Comment

-When reviewing the confusion matrices of the supplementary material, it is not understood why for some models the matrix is reported with 1063 positive cases or other models with 101 positive cases and others with 119.

R/ Thank you for your observation. In our previous analysis, some results were indeed derived using out-of-bag (OOB) samples, which led to discrepancies in the confusion matrices, such as the varying number of positive cases reported.

To address this issue, we have re-evaluated all models using a balanced sample. All reported results are now based on the balanced test set using cross-validation. This ensures consistency and accuracy in the confusion matrices and other performance metrics.

Other suggestions:

The manual labels were made by a single specialist. At least two specialists are suggested to perform concordance measurements.

R/ Thank you for your comment. Due to practical constraints, it was not feasible to have the review conducted by two specialists. However, to ensure the correct classification of the clinical records, we developed a list of predefined criteria corresponding to graft loss. These criteria were meticulously established in consultation with clinical experts and followed strictly by the reviewer to ensure the accuracy and reliability of the classification process.

We have added the following clarification in the manuscript:

“These criteria were meticulously established in consultation with clinical experts to ensure accuracy and reliability in the classification process. This collaborative approach with clinical experts provided a robust framework for the manual review, enhancing the validity of the findings”.

---

## [Decision Letter · Decision Letter 1]

18 Dec 2024

PONE-D-24-30503R1Using Natural Language Processing and Machine Learning to classify the status of kidney allograft in Electronic Medical Records written in SpanishPLOS ONE

Dear Dr. García-Lopez,

Thank you for submitting your manuscript to PLOS ONE. After careful consideration, we feel that it has merit but does not fully meet PLOS ONE’s publication criteria as it currently stands. Therefore, we invite you to submit a revised version of the manuscript that addresses the points raised during the review process.

We look forward to receiving your revised manuscript.

Kind regards,

Abdulaziz T. Bako, MBBS; MPH; PhD

Academic Editor

PLOS ONE

Reviewers' comments:

Reviewer's Responses to Questions

**Comments to the Author**

1. If the authors have adequately addressed your comments raised in a previous round of review and you feel that this manuscript is now acceptable for publication, you may indicate that here to bypass the “Comments to the Author” section, enter your conflict of interest statement in the “Confidential to Editor” section, and submit your "Accept" recommendation.

Reviewer #1: (No Response)

Reviewer #2: (No Response)

Reviewer #3: All comments have been addressed

2. Is the manuscript technically sound, and do the data support the conclusions?

Reviewer #1: Partly

Reviewer #2: Partly

Reviewer #3: Yes

3. Has the statistical analysis been performed appropriately and rigorously? 

Reviewer #1: Yes

Reviewer #2: No

Reviewer #3: Yes

4. Have the authors made all data underlying the findings in their manuscript fully available?

Reviewer #1: Yes

Reviewer #2: No

Reviewer #3: Yes

5. Is the manuscript presented in an intelligible fashion and written in standard English?

Reviewer #1: Yes

Reviewer #2: Yes

Reviewer #3: Yes

6. Review Comments to the Author

Reviewer #1: The authors improved the manuscript according to the suggestions in the first review. However, the following major issues have been addressed:

In the methods section, you explain how you balanced the dataset. That is a necessary step in training the models. However, testing the models in the balanced dataset differs from a real scenario. Please train in a balanced dataset and test in an unbalanced test set. If you cannot test in an unbalanced scenario, you can describe this fact as a limitation of the study.

When you work with machine learning models, splitting the dataset into a train set and a test set is necessary. You can perform the cross-validation in the training set, but you must also report the performance metrics in the test set. It is better if the test set is unbalanced, although the train set is balanced. You must declare this in the limitations if you can follow these steps. Additionally, it is necessary to declare the hyperparameters used in the machine learning models. If you perform a hyperparameter optimization, you must state the methods used.

In the key findings section, avoid using the word "demonstrated." This preliminary study needs other validations to be "demonstrated." Instead, you can use words like "show," "suggest," and others.

Reviewer #2: General comments

In this paper, the authors developed several classification models using features extracted from unstructured clinical texts to predict the kidney allograft. The authors did a lot of manual work to implement their models. However, there are some limitations in the current manuscript.

Major comments

1.The innovation of methodology seems weak. The authors employed common methods to extract features from clinical text, select features, and develop prediction models.

2.If the “Text mining search” procedure leads to some false negative samples? Some positive samples’ clinical text may not contain the listed keywords, which will be neglected by the following manual review.

3.I noticed that some medical abbreviations like “k” and “h” in Table A.4 are the same as the custom stopwords. As the authors first removed the custom stopwords and then expanded the medical abbreviations, how to ensure that the medical abbreviations are not deleted by mistake.

4.Please indicate how the feature selection was conducted. The current study did not specify whether the feature selection was performed on the entire dataset or on the training set. Conducting on the entire dataset may lead to test data leakage.

5.The authors should specify how they split training, validation, and test sets when using 10-fold cross validation. In my opinion, the current study just selected several groups of hyperparameters first. Then, the authors trained the models with these hyperparameters and tested the models using a 10-fold CV. If it is true, the current results are just validation results but not test results, which can not reflect the true generalization ability of the models. The authors should use the 10-fold CV first. During each fold iteration, the authors should split some samples from the 9-fold data as the validation set and leave the remaining samples of the 9-fold data as the training set. After that, they should conduct feature selection on the training set, train the models using the selected features and different hyperparameters, validate the trained models on the validation set to select the best hyperparameters, and finally, test the selected models on the test set.

6.The current study did not compare the developed models with SOTA models, such as BERT or ChatGPT. Besides, they should also compare with the Rule-based method. The results in Table 3 are validation results with different hyperparameters, which is inappropriate.

Reviewer #3: (No Response)

7. PLOS authors have the option to publish the peer review history of their article (what does this mean? ). If published, this will include your full peer review and any attached files.

**Do you want your identity to be public for this peer review?** For information about this choice, including consent withdrawal, please see our Privacy Policy .

Reviewer #1: **Yes: ** Andrés Orozco-Duque

Reviewer #2: No

Reviewer #3: No

---

## [Author Response · Author response to Decision Letter 2]

3 Feb 2025

'Response to Reviewers'

Comments to the Author

Review Comments to the Author

Reviewer #1:

The authors improved the manuscript according to the suggestions in the first review. However, the following major issues have been addressed:

In the methods section, you explain how you balanced the dataset. That is a necessary step in training the models. However, testing the models in the balanced dataset differs from a real scenario. Please train in a balanced dataset and test in an unbalanced test set. If you cannot test in an unbalanced scenario, you can describe this fact as a limitation of the study.

R/ Thank you for your valuable comments. We have accepted your suggestion and made the requested changes. The model has now been trained on a balanced dataset and tested on an unbalanced test set. The results of these tests have been included in the Results section of the revised manuscript (Table 3). Additionally, we have updated the Methods section to explain this process in detail.

When you work with machine learning models, splitting the dataset into a train set and a test set is necessary. You can perform the cross-validation in the training set, but you must also report the performance metrics in the test set. It is better if the test set is unbalanced, although the train set is balanced. You must declare this in the limitations if you can follow these steps.

R/ Thank you for your insightful comments. We have accepted your suggestion and made the requested changes. The following modifications were implemented:

1. The dataset was initially partitioned using an 80/20 split, where 80% of the data was allocated for training and validation, and the remaining 20% was reserved for testing.

2. The training and validation set underwent a 10-fold cross-validation process. In this method, the data was divided into 10 equal folds, with 9 folds used for training and 1 fold used for validation in each iteration. This cross-validation process was repeated 10 times, ensuring that each fold was used as the validation set exactly once.

3. The final model performance was evaluated on the test set, which comprised the remaining 20% of the data.

These changes have been detailed in the Methods section, and the performance metrics have been reported in the Results section. As suggested, the test set was unbalanced to better reflect real-world scenarios.

Additionally, it is necessary to declare the hyperparameters used in the machine learning models. If you perform a hyperparameter optimization, you must state the methods used.

R/ Thank you for your valuable comments. We attempted to use hyperparameter optimization; however, it required significant computational resources, which we do not have. As a result, each model reported has the best combination of hyperparameters tested manually. We have declared the in the Methods section that hyperparameter optimization techniques were not used and, in the limitations section, we declared “Furthermore, hyperparameter optimization techniques, which could significantly enhance model performance, were not utilized due to the high computational demands, making it unfeasible for this particular study.”

We appreciate your observation, as it has helped improve the clarity and transparency of our study.

In the key findings section, avoid using the word "demonstrated." This preliminary study needs other validations to be "demonstrated." Instead, you can use words like "show," "suggest," and others.

R/ Thank you for your feedback. We have made the suggested changes in the key findings section. The word "demonstrated" has been replaced.

Reviewer #2:

General comments

In this paper, the authors developed several classification models using features extracted from unstructured clinical texts to predict the kidney allograft. The authors did a lot of manual work to implement their models. However, there are some limitations in the current manuscript.

Major comments

1.The innovation of methodology seems weak. The authors employed common methods to extract features from clinical text, select features, and develop prediction models.

Thank you for your feedback. You are correct that the methods we used, such as information extraction through NLP, variable selection with LASSO regression, and the construction of prediction models, are well-established and widely supported in the scientific literature. However, the contribution of this study lies in applying these widely used techniques to address a persistent problem: the management of abundant unstructured clinical data and the manual work it entails. Additionally, this study makes a significant contribution to the field of transplants, which lacks published research on this topic, and it is the first study in the field working with text written in Spanish.

Furthermore, we recognize that this is a first step, and future work could leverage Large Language Models (LLMs) and other developments in artificial intelligence to further enhance the analysis and classification capabilities.

2.If the “Text mining search” procedure leads to some false negative samples? Some positive samples’ clinical text may not contain the listed keywords, which will be neglected by the following manual review.

R/ It is possible that the text mining search did not include some keywords. However, we made every effort to minimize this risk. The search criteria were clinically reviewed, and upon examining the texts, many patients with graft loss contain more than one indicative word in their texts. Therefore, we expect that at this stage the loss of cases will be minimal.

3.I noticed that some medical abbreviations like “k” and “h” in Table A.4 are the same as the custom stopwords. As the authors first removed the custom stopwords and then expanded the medical abbreviations, how to ensure that the medical abbreviations are not deleted by mistake.

R/Thank you for noticing this. In the Methods section, we clarified that common stopwords were removed before expanding medical abbreviations. Additionally, custom stopwords were removed at the end of the preprocessing phase, ensuring that medical abbreviations were not mistakenly deleted.

4.Please indicate how the feature selection was conducted. The current study did not specify whether the feature selection was performed on the entire dataset or on the training set. Conducting on the entire dataset may lead to test data leakage.

R/ Thank you for your observation. Feature selection was conducted on the training set, as specified in the Methods section. We used two techniques: bivariate analysis with a p-value threshold of ≤ 0.01 for a statistically significant association with graft loss, and LASSO (least absolute shrinkage and selection operator) to select the highest coefficients.

5.The authors should specify how they split training, validation, and test sets when using 10-fold cross validation. In my opinion, the current study just selected several groups of hyperparameters first. Then, the authors trained the models with these hyperparameters and tested the models using a 10-fold CV. If it is true, the current results are just validation results but not test results, which can not reflect the true generalization ability of the models. The authors should use the 10-fold CV first. During each fold iteration, the authors should split some samples from the 9-fold data as the validation set and leave the remaining samples of the 9-fold data as the training set. After that, they should conduct feature selection on the training set, train the models using the selected features and different hyperparameters, validate the trained models on the validation set to select the best hyperparameters, and finally, test the selected models on the test set.

R/ Thank you for your insightful comments. We have accepted your suggestion and made the requested changes. The following modifications were implemented:

1. The dataset was initially partitioned using an 80/20 split, where 80% of the data was allocated for training and validation, and the remaining 20% was reserved for testing.

2. The training and validation set underwent a 10-fold cross-validation process. In this method, the data was divided into 10 equal folds, with 9 folds used for training and 1 fold used for validation in each iteration. This cross-validation process was repeated 10 times, ensuring that each fold was used as the validation set exactly once.

3. Feature selection was performed in the training set.

4. The final model performance was evaluated on the test set, which comprised the remaining 20% of the data.

These changes have been detailed in the Methods section, and the performance metrics have been reported in the Results section. As suggested, the test set was unbalanced to better reflect real-world scenarios.

6.The current study did not compare the developed models with SOTA models, such as BERT or ChatGPT. Besides, they should also compare with the Rule-based method.

R/ Thank you for your valuable feedback. We acknowledge that the current study did not compare the developed models with state-of-the-art (SOTA) models such as BERT or ChatGPT. Future work will include these comparisons to provide a more comprehensive evaluation.

In the manuscript, we recognized that the use of large language models would add value and will be considered for future work. Conducting these comparisons may be beyond the scope of this study, given that it was initiated in 2023 when the use of these models was still emerging. Nonetheless, we believe that our study contributes valuable insights into the use of natural language processing tools and serves as a first step towards more complex models.

The results in Table 3 are validation results with different hyperparameters, which is inappropriate.

R/ Thank you for your observation. We have made the suggested changes by training the models on 9-folds of the cross-validation, validated on 1-fold with different hyperparameters, and tested on the imbalanced test set to evaluate the model in a real-world scenario. We selected the top three algorithms from each class—Logistic Regression, Random Forest, and Neural Network—for comparison. As a result, the outcomes in Table 3 have been updated accordingly.

Reviewer #3: (No Response)

---

## [Decision Letter · Decision Letter 2]

19 Feb 2025

PONE-D-24-30503R2Using Natural Language Processing and Machine Learning to classify the status of kidney allograft in Electronic Medical Records written in SpanishPLOS ONE

Dear Dr. García-Lopez,

Thank you for submitting your manuscript to PLOS ONE. After careful consideration, we feel that it has merit but does not fully meet PLOS ONE’s publication criteria as it currently stands. Therefore, we invite you to submit a revised version of the manuscript that addresses the points raised during the review process.

We look forward to receiving your revised manuscript.

Kind regards,

Abdulaziz T. Bako, MBBS; MPH; PhD

Academic Editor

PLOS ONE

Journal Requirements:

Reviewers' comments:

Reviewer's Responses to Questions

**Comments to the Author**

1. If the authors have adequately addressed your comments raised in a previous round of review and you feel that this manuscript is now acceptable for publication, you may indicate that here to bypass the “Comments to the Author” section, enter your conflict of interest statement in the “Confidential to Editor” section, and submit your "Accept" recommendation.

Reviewer #1: All comments have been addressed

Reviewer #2: (No Response)

Reviewer #3: All comments have been addressed

2. Is the manuscript technically sound, and do the data support the conclusions?

Reviewer #1: Yes

Reviewer #2: Yes

Reviewer #3: Yes

3. Has the statistical analysis been performed appropriately and rigorously? 

Reviewer #1: Yes

Reviewer #2: Yes

Reviewer #3: Yes

4. Have the authors made all data underlying the findings in their manuscript fully available?

Reviewer #1: Yes

Reviewer #2: No

Reviewer #3: Yes

5. Is the manuscript presented in an intelligible fashion and written in standard English?

Reviewer #1: Yes

Reviewer #2: Yes

Reviewer #3: Yes

6. Review Comments to the Author

Reviewer #1: The authors have addressed the suggestions in the previous review, incorporating the proposed revisions. The modifications significantly improve the paper's clarity and coherence.

Some minor issues:

1. Please include the link to the dataset repository in the "Data source and study population" section to make it available to readers.

2. In the "Comparison with other models" section, the authors claim: "When comparing our NLP algorithms to other state-of-the-art techniques in similar domains, our models showed considerable advantages in terms of discrimination, calibration, accuracy, and sensitivity." And, in key findings the authors wrote: "the Random Forest model outperformed the others achieving an AUC of 0.98 and an F1 score of 0.65, indicating strong discrimination between EMRs that recorded graft loss and those that did not, as well as a reasonable balance between precision and recall." However, discussing the implication of the low PPV (0.56) and the modest sensitivity is necessary. Although the results reach high specificity (0.99) and moderate F1 score (0.65), the PPV of 0.56 tells us that there are still many predicted positives that actually are not. In highly imbalanced problems, the PPV often suffers.

3. The abstract's result section focuses mainly on AUC results in the validation dataset. It is more important to discuss the results in the test set. Consider that a high ROC AUC and accuracy in highly imbalanced datasets can be misleading because the majority class can strongly influence the metric. Then, it is better to show the F1-score, sensitivity, and specificity results. This recommendation is an important fact in the discussion section, too.

Reviewer #2: Thanks for considering my comments. For my fifth comment, the authors conducted their experiment in a different way, but it's OK. One more thing I need to verify is that the authors performed the 10-fold cross-validation on the training-validation set to make the hyperparameter selection, then re-train the model using the selected hyperparameters on the entire training-validation set as the final model and finally tested the final model on the test set?

Reviewer #3: (No Response)

7. PLOS authors have the option to publish the peer review history of their article (what does this mean? ). If published, this will include your full peer review and any attached files.

**Do you want your identity to be public for this peer review?** For information about this choice, including consent withdrawal, please see our Privacy Policy .

Reviewer #1: **Yes: ** Andrés Orozco-Duque

Reviewer #2: No

Reviewer #3: No

---

## [Author Response · Author response to Decision Letter 3]

27 Feb 2025

Journal Requirements:

R/ We have carefully reviewed our reference list and confirm that it is now complete and correct. Additionally, we have verified each cited article and confirm that none have been retracted.

Reviewers' comments:

Reviewer #1: The authors have addressed the suggestions in the previous review, incorporating the proposed revisions. The modifications significantly improve the paper's clarity and coherence.

Some minor issues:

1. Please include the link to the dataset repository in the "Data source and study population" section to make it available to readers.

R/We have included the reference to the dataset repository in the "Data Source and Study Population" section, and it has been properly cited.

Thank you for your suggestion.

2. In the "Comparison with other models" section, the authors claim: "When comparing our NLP algorithms to other state-of-the-art techniques in similar domains, our models showed considerable advantages in terms of discrimination, calibration, accuracy, and sensitivity." And, in key findings the authors wrote: "the Random Forest model outperformed the others achieving an AUC of 0.98 and an F1 score of 0.65, indicating strong discrimination between EMRs that recorded graft loss and those that did not, as well as a reasonable balance between precision and recall." However, discussing the implication of the low PPV (0.56) and the modest sensitivity is necessary. Although the results reach high specificity (0.99) and moderate F1 score (0.65), the PPV of 0.56 tells us that there are still many predicted positives that actually are not. In highly imbalanced problems, the PPV often suffers.

R/ We appreciate your feedback. In response to your comment, we have made the necessary clarification in the discussion section. We have added the following statement:

Key findings:

"Notably, the model showed a low PPV (0.56) and modest sensitivity (0.76), indicating that a significant number of predicted positives are false positives, which implies additional manual work to verify these cases."

Comparison with other models:

“However, while these models can be useful in clinical practice, manual work may still be required due to the high number of false positives”.

This addition addresses the implications of the low PPV and modest sensitivity, highlighting the need for further manual verification in practical applications

3. The abstract's result section focuses mainly on AUC results in the validation dataset. It is more important to discuss the results in the test set. Consider that a high ROC AUC and accuracy in highly imbalanced datasets can be misleading because the majority class can strongly influence the metric. Then, it is better to show the F1-score, sensitivity, and specificity results. This recommendation is an important fact in the discussion section, too.

R/ We have adjusted the abstract in the results and conclusion sections following your suggestion. The abstract now focuses on the results from the test set, emphasizing the importance of the F1-score, sensitivity, and specificity rather than just the AUC. 

The discussion section also included this recommendation.

Reviewer #2: Thanks for considering my comments. For my fifth comment, the authors conducted their experiment in a different way, but it's OK. One more thing I need to verify is that the authors performed the 10-fold cross-validation on the training-validation set to make the hyperparameter selection, then re-train the model using the selected hyperparameters on the entire training-validation set as the final model and finally tested the final model on the test set?

R/ Thank you for your comment. Indeed, we first selected the hyperparameters using the training-validation set and trained the models with these chosen hyperparameters on the same training-validation set. Based on the results of the 10-fold cross-validation, we then tested the models on the test set. The final selected and tested hyperparameters for both validation and test sets are detailed in the supplementary code file, where we retained the best-performing hyperparameters.

Reviewer #3: (No Response)

---

## [Decision Letter · Decision Letter 3]

24 Mar 2025

Using Natural Language Processing and Machine Learning to classify the status of kidney allograft in Electronic Medical Records written in Spanish

PONE-D-24-30503R3

Dear Dr. Garcia-Lopez,

We’re pleased to inform you that your manuscript has been judged scientifically suitable for publication and will be formally accepted for publication once it meets all outstanding technical requirements.

Kind regards,

Abdulaziz T. Bako, MBBS; MPH; PhD

Academic Editor

PLOS ONE

Additional Editor Comments (optional):

Reviewers' comments:

Reviewer's Responses to Questions

**Comments to the Author**

1. If the authors have adequately addressed your comments raised in a previous round of review and you feel that this manuscript is now acceptable for publication, you may indicate that here to bypass the “Comments to the Author” section, enter your conflict of interest statement in the “Confidential to Editor” section, and submit your "Accept" recommendation.

Reviewer #1: All comments have been addressed

Reviewer #2: All comments have been addressed

2. Is the manuscript technically sound, and do the data support the conclusions?

Reviewer #1: Yes

Reviewer #2: Yes

3. Has the statistical analysis been performed appropriately and rigorously? 

Reviewer #1: Yes

Reviewer #2: Yes

4. Have the authors made all data underlying the findings in their manuscript fully available?

Reviewer #1: Yes

Reviewer #2: Yes

5. Is the manuscript presented in an intelligible fashion and written in standard English?

Reviewer #1: Yes

Reviewer #2: Yes

6. Review Comments to the Author

Reviewer #1: The authors have addressed all the concerns highlighted in previous review rounds, and the manuscript has undergone substantial improvements. I consider that the manuscript meets the required recommendation to be accepted.

Reviewer #2: Thanks for addressing my concern. The authors made a good work. I think the current paper is ready for acceptance.

7. PLOS authors have the option to publish the peer review history of their article (what does this mean? ). If published, this will include your full peer review and any attached files.

**Do you want your identity to be public for this peer review?** For information about this choice, including consent withdrawal, please see our Privacy Policy .

Reviewer #1: No

Reviewer #2: No

---

## [Editor Report · Acceptance letter]

PONE-D-24-30503R3

PLOS ONE

Dear Dr. García-Lopez,

I'm pleased to inform you that your manuscript has been deemed suitable for publication in PLOS ONE. Congratulations! Your manuscript is now being handed over to our production team.

Kind regards,

on behalf of

Dr. Abdulaziz T. Bako

Academic Editor

PLOS ONE